# Numerical Analysis of Axial Cyclic Behavior of FRP Retrofitted CHS Joints

**DOI:** 10.3390/ma14030648

**Published:** 2021-01-31

**Authors:** Mohammad Alembagheri, Maria Rashidi, Amin Yazdi, Bijan Samali

**Affiliations:** 1Centre for Infrastructure Engineering, School of Engineering, Western Sydney University, Penrith 2749, Australia; m.rashidi@westernsydney.edu.au (M.R.); b.samali@westernsydney.edu.au (B.S.); 2Department of Civil and Environmental Engineering, Tarbiat Modares University, Tehran 11141615, Iran; 3Department of Civil and Environmental Engineering, Amirkabir University of Technology, Tehran 11154585, Iran; aminyazdi@aut.ac.ir

**Keywords:** tubular T-joints, ultimate strength, cyclic loading, welding, FRP strengthening, buckling

## Abstract

This paper aims to numerically investigate the cyclic behavior of retrofitted and non-retrofitted circular hollow section (CHS) T-joints under axial loading. Different joints with varying ratios of brace to chord radius are studied. The effects of welding process on buckling instability of the joints in compression and the plastic failure in tension are considered. The finite element method is employed for numerical analysis, and the SAC protocol is considered as cyclic loading scheme. The CHS joints are retrofitted with different numbers of Fiber Reinforced Polymer (FRP) layers with varying orientation. The results show that the welding process significantly increases the plastic failure potential. The chord ovalization is the dominant common buckling mode under the compression load. However, it is possible to increase the energy dissipation of the joints by utilizing FRP composite through changing the buckling mode to the brace overall buckling.

## 1. Introduction

Circular hollow sections (CHS) have been widely used in steel structures in the last decades. This is because of their appropriate properties such as low weight and proper performance under cyclic loadings. They lead to the broad use of CHS for structures located in active seismic regions, specifically in onshore and offshore structures which are subjected to sea waves cyclic load. However, because of destructive environmental effects resulting in corrosion of structural components or even bad designing and incorrect utilization, they may exhibit poor performance. Hence, it may be required to use retrofitting methods to retrieve these or other similar deficiencies and improve the structure performance under cyclic loading. There are several methods which can be used to retrofit components of steel structures, specifically their joints. The conventional method of repairing or rehabilitating is cutting out and replacing the poor steel sections. This kind of retrofitting is usually bulky, heavy, difficult to fix, and prone to corrosion and fatigue [1,2,3,4]. The alternative option is wrapping the steel sections by Fiber Reinforce Polymer (FRP) composites. The FRP patches properly resist against fire and environmental corrosion; they also show good mechanical performance [5,6,7,8]. There is large demand for investigating retrofitted steel structures using FRP materials [9,10,11,12,13,14,15].

The performance of the joints in steel structures is crucial for the whole structural performance. Most vulnerability of steel structures is caused by the joints failure in various modes, thus importance of investigation on steel joints is remarkable. The CHS or the tubular T-joints mainly consist of a main member named as chord and an intersecting member named as brace (Figure 1). Several failure modes can happen for tubular steel T-joints under common or extreme load cases. They mostly include cracking of the welding zones in the intersection of brace and chord surface, plastic failure, and local or overall buckling of brace or chord. One of the main loadings of these joints is axial loading applied at the end of the brace in tension or compression. The mentioned failure modes may happen under this kind of axial loading, especially if the load is applied in a cyclic manner [16]. The cyclic behavior of the CHS joints under lateral loading applied to their brace and their monotonic behavior under axial loading have been experimentally and numerically investigated [17,18]. Batuwitage et al. showed that the chord ovalization may happen under compression load, and it is possible to improve load bearing capacity of CHS T-joints by FRP jackets [14]. The FRP can decrease stresses and deformations in chord ovalization mode [14]. Wang and Chen studied eight CHS T-joints; four of them were investigated under cyclic axial load, and the others were evaluated under cyclic in-plane bending. Two failure modes were observed and, generally, the results revealed the weld cracking in tension and the chord plastic failure in compression. Additionally, the load bearing capacity of the joints under monotonic loading was compared with cyclic ones [16]. Numerical models were adopted to predict performance of CHS joints under severe earthquake excitations [19]. Chen et al. used the finite element (FE) simulation for debonding failures in FRP-strengthened concrete beams. They examined the effectiveness of using a dynamic analysis approach in FE simulations, in which debonding failure was treated as a dynamic problem and solved using an appropriate time integration method. They presented numerical results to show that an appropriate dynamic approach effectively overcomes the convergence problem and provides accurate predictions of test results [20]. Experimental and numerical investigations were conducted on un-stiffened space tubular joints. Two full scale specimens under monotonic and cyclic quasi-static loading were tested, and four multi-planar CHS joints were experimentally analyzed under both kinds of monotonic and cyclic load. The results showed the local buckling in segments of the CHS joints that may lead to energy dissipating mechanism [21]. Ombres and Verre investigated the bond between a composite strengthening system consisting of steel textiles embedded into an inorganic matrix (steel reinforced grout, SRG) and the concrete substrate. An experimental investigation was carried out on medium density SRG specimens; direct shear tests were conducted on 20 specimens to analyze the effect of the bond length and the age of the composite strip on the SRG-to-concrete bond behavior. They developed a finite element model to replicate the behavior of SRG strips [22]. Concrete filled steel CHS T-joints were investigated under in-plane cyclic bending load [23]. For this purpose, two specimens were tested under cyclic bending load applied to the brace end, while constant compression load was applied to the concrete filled chord. It has been revealed that the prevailing failure mode was punching shear failure on chord surface. An investigation on four CHS T-joints under axial cyclic loading was carried out where the chord was filled by concrete. Experimental evaluation was based on investigation of energy dissipation, ductility, stiffness, and strength deterioration [24]. The results showed that the most governing failure mode is crack initiation on the chord surface and convex deformation at the intersection of the brace and the chord. In spite of this research, investigating cyclic behavior of the CHS joints under axial loading is rare and deserves more investigation.

The present research aims to investigate the performance of retrofitted and non-retrofitted CHS T-joints under axial cyclic loading applied to the end of the brace. Various joint configurations with varying ratios of brace to chord radius are studied. The SAC protocol is employed as loading scenario. The joints are numerically modeled using the finite element method, and the most dominant failure modes are determined. The effect of welding process is considered to obtain realistic results, especially in the plastic failure mode under tension load. The welding effects consist of residual stresses and deformations which appear by heat distribution in the process of welding that connects the brace to the chord surface. The joints are retrofitted using different FRP patterns wrapped around the brace and the chord. The number of the FRP layers and their orientation are assessed in a parametric analysis to find the most suitable ones. 

## 2. Structural Model

The schematic representation of non-retrofitted CHS T-joints and their geometric features are shown in Figure 1. Three different configurations are investigated in this study; their geometric properties are listed in Table 1. These geometric configurations have been selected among many other models that were primarily studied to show various buckling modes. One of the most influencing geometric parameters is the ratio of the brace radius to the chord radius (*β* factor). As is apparent in Table 1, each considered T-joint has a unique *β* factor. In this table, L represents the brace length that affects the buckling response of the joints. The chord length is considered 3000 mm constant in all models. The welding process connecting the brace to the chord is assumed as one circumferential pass. The initial temperature of the steel is assumed to be 21 °C, while the temperature of the welding is considered as 1500 °C. This high difference between the temperatures causes the residual stresses within the steel after the welding process is completed and the joints are cooled.

Because the welding process is modeled, it is required to perform a coupled thermal-displacement analysis in the numerical model. To do this, the temperature-dependent mechanical properties of the steel should be determined as shown in Table 2. The same steel properties are used for both chord and brace. The steel material is assumed to be homogeneous and isotropic. The nonlinear behavior of the steel is considered as elastic-perfectly plastic. The steel Poisson’s ratio is considered constant as 0.3.

The SAC protocol [26] is used as displacement-control cyclic loading scheme. It includes 32 cycles of loading that are axially applied along the brace, as shown in Figure 1. The cyclic loading is applied to the model after the welding process is completed and the joints are cooled. The SAC protocol and its amplitudes are shown in Figure 2. The amplitudes of the SAC protocol are based on radian; the equivalent displacement amplitude is obtained as multiplication of the amplitude and half of the brace length. This protocol contains 32 cycles of loading. 

The T-joints are retrofitted using FRP jackets, which are wrapped around the chord and the brace. Opposite to the steel, the FRP material shows anisotropic behavior in its matrix and fiber directions. The length of the FRP jackets is considered as one-third of the chord length. The FRP material is assumed as carbon epoxy type with properties listed in Table 3 [27]. In this table, E_1_ and E_2_ are the Young’s modulus in fiber and matrix (perpendicular to the fiber) directions, respectively. G_12_, G_23_, G_13,_ and υ_12_ are the shear modulus and the Poisson’s ratio of composite material, respectively, in three main orthogonal directions. Different numbers of layers and layer orientations are investigated to study the effects of FRP retrofitting. They include three different layers for both the chord and the brace patches, which are 4, 8, and 12 layers with ply distribution of (8)s, (0/90)2s, and (0/90)3s, respectively. The FRP patches are considered with perfect bond, and the loading is applied statically. 

## 3. Finite Element Model and Verification

The joints are modeled using the finite element method in three-dimensional space. The chord and the brace are discretized using linear hexahedral solid elements. The finite element mesh of the J1 model is shown in Figure 3. To accurately analyze the welding process, finer mesh is used in the welding zone. The same mesh density is used for the other joints. For FRP patches, linear quadrilateral shell elements are employed. The adopted meshes are refined such that they have negligible effects on the obtained results. The FRP patches are fully bonded to the steel tubes. The SAC protocol is applied as displacement-control loading at the end of the brace along the brace longitudinal axis, while it is laterally constrained. The pinned condition is assigned for both ends of the chord. Rotational degree of freedom about chord longitudinal axis is constrained to prevent chord rotational instability. 

In all three joint models, the welding process consists of one clockwise circumferential pass, which has 72 beads. The welding pass speed is assumed 80 mm/min. Maximum imperfection in all models is 0.001 mm, and imperfection pattern is considered as the first governing buckling mode.

The numerical model is verified using a typical CHS T-joint under static monotonic compression load, which was carried out numerically and experimentally by Lesani et al. [21]. For more details about the model, one can refer to [21]. The obtained results are in good agreement with the mentioned study, as shown in Figure 4. For verification analysis, loads are applied to the brace end in the same way as in the current cyclic investigation. 

## 4. Investigating the Welding Process

As it was stated, the welding process is modeled before applying the cyclic axial load to the brace end. Figure 5 shows the contours of residual stress and equivalent plastic strain for J1 and J2 models after the welding process analysis. From this figure, the welding process causes the yielding of steel and residual stresses. The plastic strain up to 0.025 can be produced locally around the welded sections because of high temperature gradients resulting from the welding process. The residual stresses more spread in the J1 model with respect to the J2 model. It is possibly because of higher section thickness of the J1 model. The impact of the welding process is further investigated in the next section. 

## 5. Results of Cyclic Axial Loading

### 5.1. Non-Retrofitted Joints

First, the impact of the welding process on the cyclic behavior of the joints is assessed by comparing the axial (vertical) displacement load hysteresis diagrams of the non-retrofitted J1 model with and without considering the welding process, shown in Figure 6. Positive values in this figure and the other similar figures show loading and displacement in tension. From this figure, the welding process decreases the stiffness and the energy dissipation of the joint. It reduces the loading capacity of the model by 12% in compression and 31% in tension. The reason for more reduction under the tension loading is that the welding expedites the plastic failure of the model in tension. The contours of equivalent plastic strain and buckled deformation at the end of the axial cyclic loading are shown in Figure 7. In both conditions, the buckling mode in compression is the chord ovalization, which results in the concentration of plastic strains around the crown point. The crown point is shown in Figure 1. The welding causes a non-symmetric buckled shape. 

The hysteresis diagrams of the non-retrofitted J2 and J3 models, considering the welding process, are shown in Figure 8. The substantial reduction in the tensile load bearing capacity of the J2 model at the last cycles shows that the tensile plastic failure is more critical than the chord ovalization buckling mode. It may decrease the compressive load bearing capacity as well. However, the tensile load bearing capacity of the J3 model is more than the compressive one. The envelope curves of the hysteresis diagrams of the non-retrofitted joints are compared in Figure 9. It is observed that the joint with higher β factor shows higher load bearing capacity even though it has lower chord thickness. 

### 5.2. Retrofitted Joints

Improvement of the joints’ cyclic behavior is attempted using the FRP sheets. They can be wrapped around the brace, the chord, or both of them. Different retrofitting schemes are first compared using eight FRP layers with symmetric distribution of (0/90)2s; it has been shown to be one of the best ply distributions for retrofit of CHS T-joints under axial compression load [28]. Composite patches lengths for the chord and the brace are considered as 500 mm and 300 mm, respectively, which are separately bonded to the chord, the brace, and both of them. The resulted hysteresis diagrams are shown in Figure 10; also shown is the envelope curve of the non-retrofitted section. It is observed that the FRP retrofitting increases the compressive load bearing capacity, but some retrofitting scenarios may decrease the final tensile load bearing capacity of the joint. Among the retrofitted joints, the lowest loading capacity belongs to the model with only chord retrofitting. It shows that, when the brace is loaded, only retrofitting of the chord would not be considerably helpful. In addition, when the chord is retrofitted, the sudden decrease (relaxation) in tensile load bearing capacity at the last cycles is observed whether the brace is retrofitted or not. However, the compressive loading capacity increases when both components are retrofitted. 

The contours of equivalent plastic strains and the buckled deformation shapes at the end of the cyclic loading are shown in Figure 11. When both chord and brace are retrofitted, the buckling mode is transmitted from the chord to the brace; it is changed to the overall brace buckling that results in the most load bearing capacity in compression. In this condition, the plastic strains are concentrated around the saddle point, whereas when one component is retrofitted, the plastic strains are more concentrated around the crown point. 

To investigate the effects of the number of plies, both chord and brace of the joints are identically retrofitted with 4, 8, and 12 FRP layers with symmetric cross-ply distributions of (0/90)s, (0/90)2s, and (0/90)3s, respectively. The envelope curves of their hysteresis load displacement diagrams are compared in Figure 12. The same length is assumed for the patches as before. The retrofitting may increase the joints stiffness with almost the same value in both tension and compression. This increase is lower for the joints with higher *β* factor. The retrofitting also increases the ultimate load bearing capacity of the joints; more layers causes more increase. However, the final load bearing capacity of the joints may decrease, especially in tension that shows un-usefulness of the FRP retrofitting against tensile plastic failure. Hence, there is a difference between the load-bearing capacities in various cycles of the cyclic loading. In general, the effects of the FRP retrofitting are more obvious for the joints with lower *β* factor. 

The load bearing capacity and the energy dissipation in each cycle of the cyclic loading for the chord and the brace-retrofitted joints are compared in Figure 13. The values are normalized with respect to the corresponding value of the non-retrofitted case and are represented as change ratios. The tensile load bearing capacity of the J1 model would increase more than 50% with respect to the non-retrofitted case until the 25th cycle of the cyclic loading, when 8- or 12-ply FRP patch is used. This load bearing capacity generally decreases during the cyclic loading. Other trends are observed for the tensile load bearing capacity of the J2 and the J3 models. No specific trend can be obtained for the J2 model, but for the J3 model, the increase is generally limited to less than 25%, even with 12-ply FRP patch. Regarding the compressive load bearing capacity, again, the lowest changes are observed for the J3 model. In the J1 and the J2 models, more than 100% increase in the compressive load bearing capacity may be obtained with 12-ply FRP retrofitting. The energy dissipation, which is the area enclosed by the load displacement hysteresis diagram, is the same between the two cases (retrofitted and non-retrofitted) in the linear region (first six cycles). After that, this value may be lower for the retrofitted joints with respect to the non-retrofitted ones up to the 12th cycle of the loading. The energy dissipation of the retrofitted J1 model would be higher after that, but for the J2 and the J3 models, the energy dissipation may be lower for the retrofitted joints when 4-ply FRP patch is used. 

## 6. Concluding Remarks

In this research, the axial cyclic behavior of circular hollow section T-joints was numerically assessed by using the finite element method. The effects of high temperature gradients caused by the welding process were considered as well. Three joints with different ratios of chord to brace radius were adopted, and the effects of FRP retrofitting on their cyclic behavior were examined. The results showed that the welding process can change initial imperfection of the joints and affect their buckling modes under compressive loading. The residual stress of the welding process can increase plastic failure potential under the tensile loading. The FRP wrapping can effectively improve the behavior of tubular T-joints under cyclic loading, especially in a compression zone. It was shown that the brace overall buckling mode has more energy dissipation than the chord ovalization buckling mode. Hence, the change of the buckling mode to the brace overall buckling by using the FRP patches would be an effective way to improve the performance of T-joints. The other conclusions of the study can be numbered as follows:Use of composite FPR patch for retrofitting the CHS T-joints can suspend the plastic failure and the chord ovalization buckling modes.Welding can decrease critical buckling load, but it can be more influential for plastic failure load.Welding process can lead to more local deformation on the chord surface.For accurate investigation of performance of CHS T-joint, it is necessary to consider the effects of the welding process, such as residual stresses and deformations.In the cyclic performance, both brace and chord must be retrofitted; by retrofitting just one of them, it can be possible that poor performance, especially in tension, is obtained.The governing buckling mode in compression load is mostly the chord ovalization.By comparing the chord ovalization and the brace overall buckling, it can be concluded that the brace overall buckling has better performance in terms of flexibility, energy dissipation, and load bearing capacity. Hence, it can be possible to use composite materials to propel the governing buckling mode to the brace overall buckling mode.Increase of the *β* factor in CHS T-joints increases the load bearing capacity in both tension and compression.

## Figures and Tables

**Figure 1 materials-14-00648-f001:**
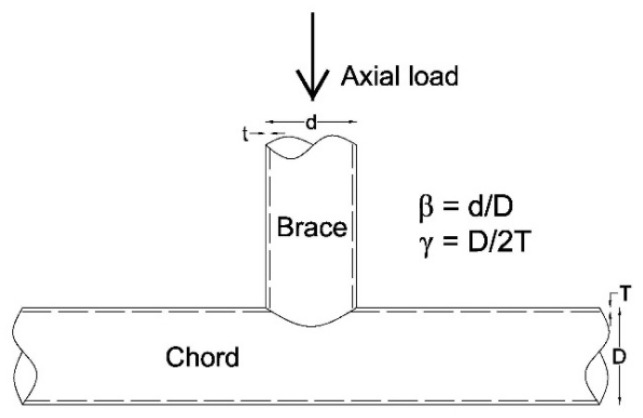
Schematic representation of tubular T-joints (circular hollow section (CHS)) [25].

**Figure 2 materials-14-00648-f002:**
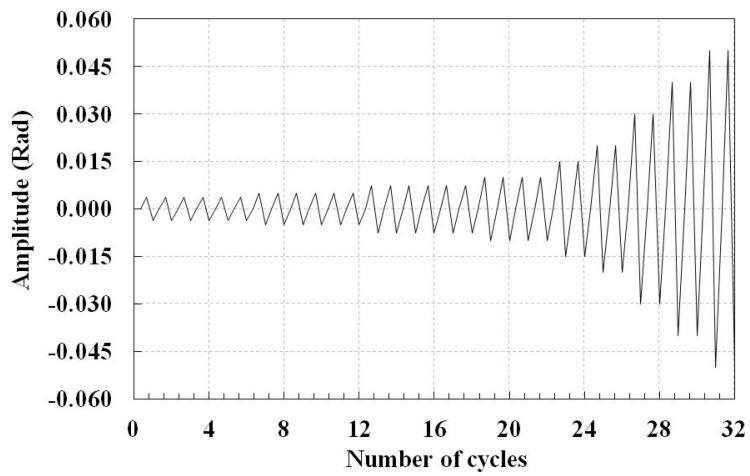
The SAC loading protocol.

**Figure 3 materials-14-00648-f003:**
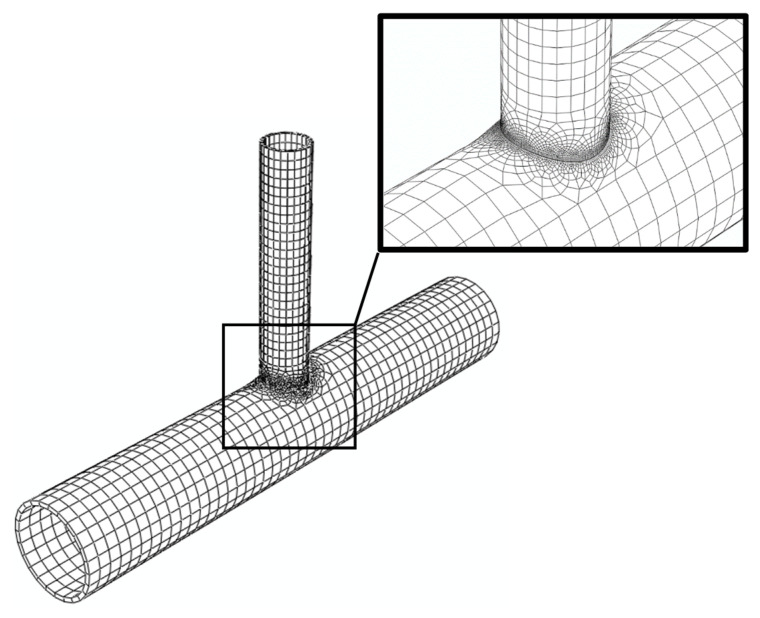
Finite element mesh of the J1 model.

**Figure 4 materials-14-00648-f004:**
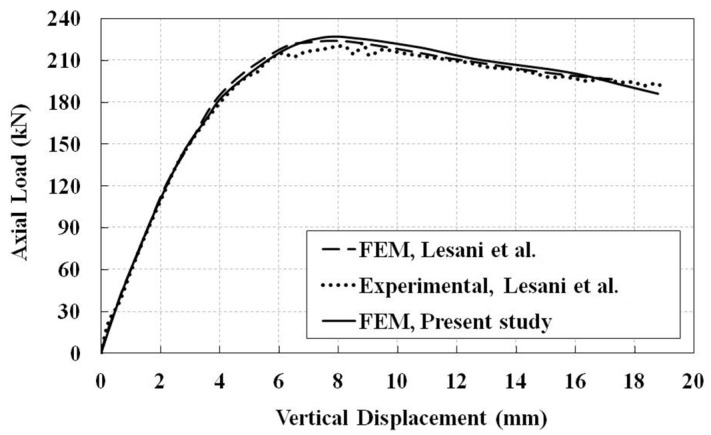
Comparison of the numerical method used in this study with Lesani et al. [10].

**Figure 5 materials-14-00648-f005:**
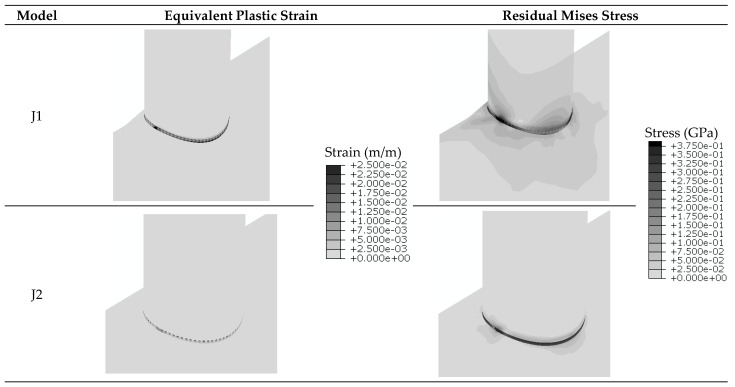
Contours of residual stress and equivalent plastic strain caused by the welding process, J1 and J2 models.

**Figure 6 materials-14-00648-f006:**
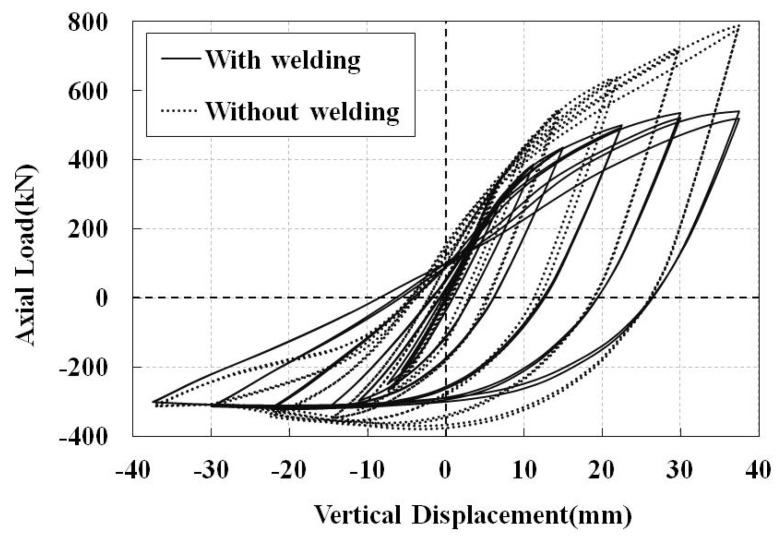
Hysteresis diagrams of the non-retrofitted J1 model with and without considering the welding process.

**Figure 7 materials-14-00648-f007:**
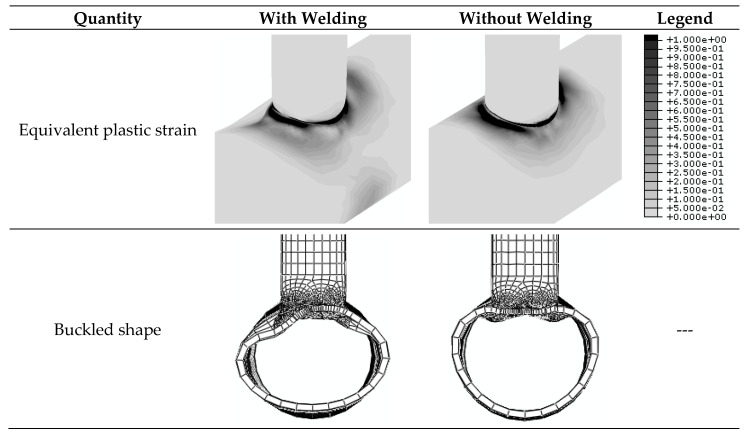
Contours of the equivalent plastic strain and the buckled deformation shape at the end of the axial cyclic loading. The J1 model with and without considering the welding process.

**Figure 8 materials-14-00648-f008:**
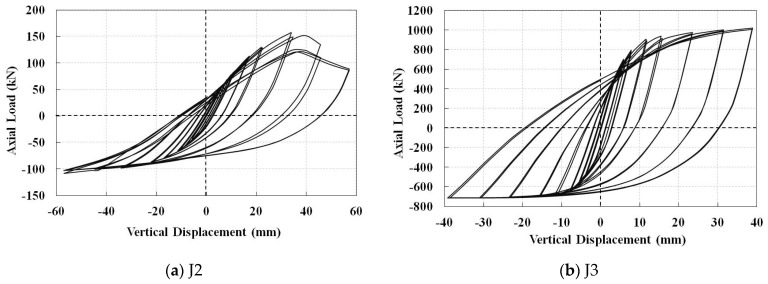
Hysteresis diagrams of the non-retrofitted J2 and J3 models, including the welding process.

**Figure 9 materials-14-00648-f009:**
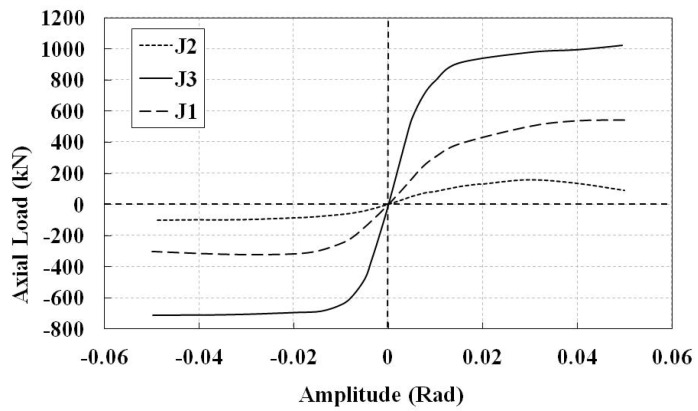
Envelope of the hysteresis diagrams. Non-retrofitted J1, J2, and J3 models including the welding process.

**Figure 10 materials-14-00648-f010:**
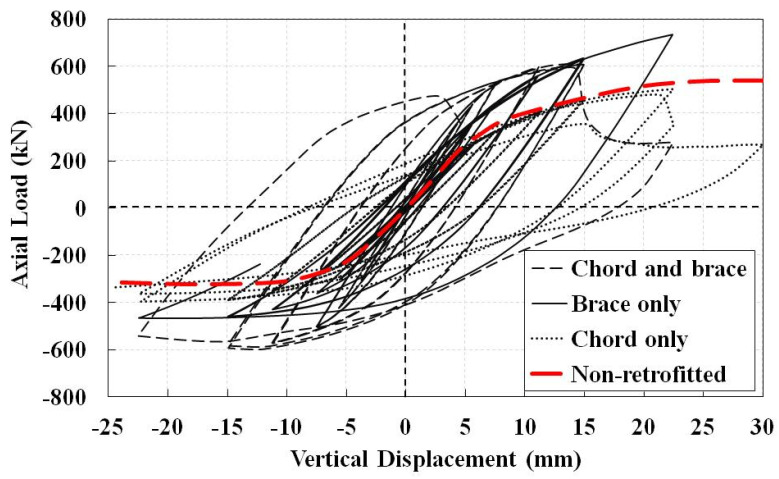
Hysteresis diagrams for different retrofitting schemes of the J1 model. Eight FRP layers with symmetric distribution of (0/90)2s.

**Figure 11 materials-14-00648-f011:**
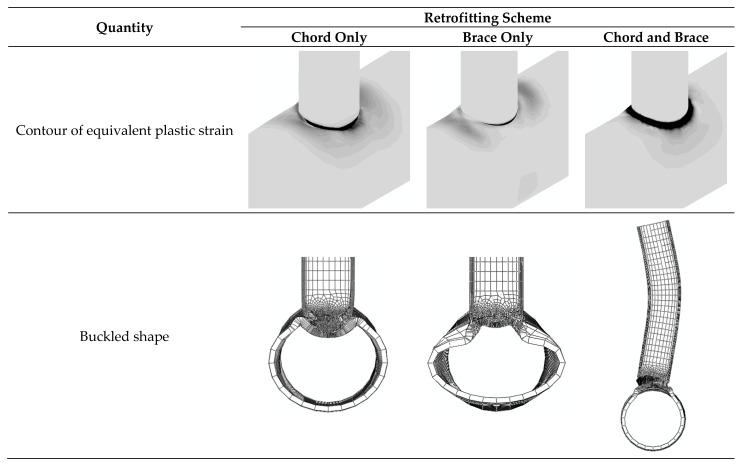
Contours of the equivalent plastic strain and the buckled deformation shapes at the end of the cyclic loading. Different retrofitting schemes. J1 model. Same legend as Figure 7.

**Figure 12 materials-14-00648-f012:**
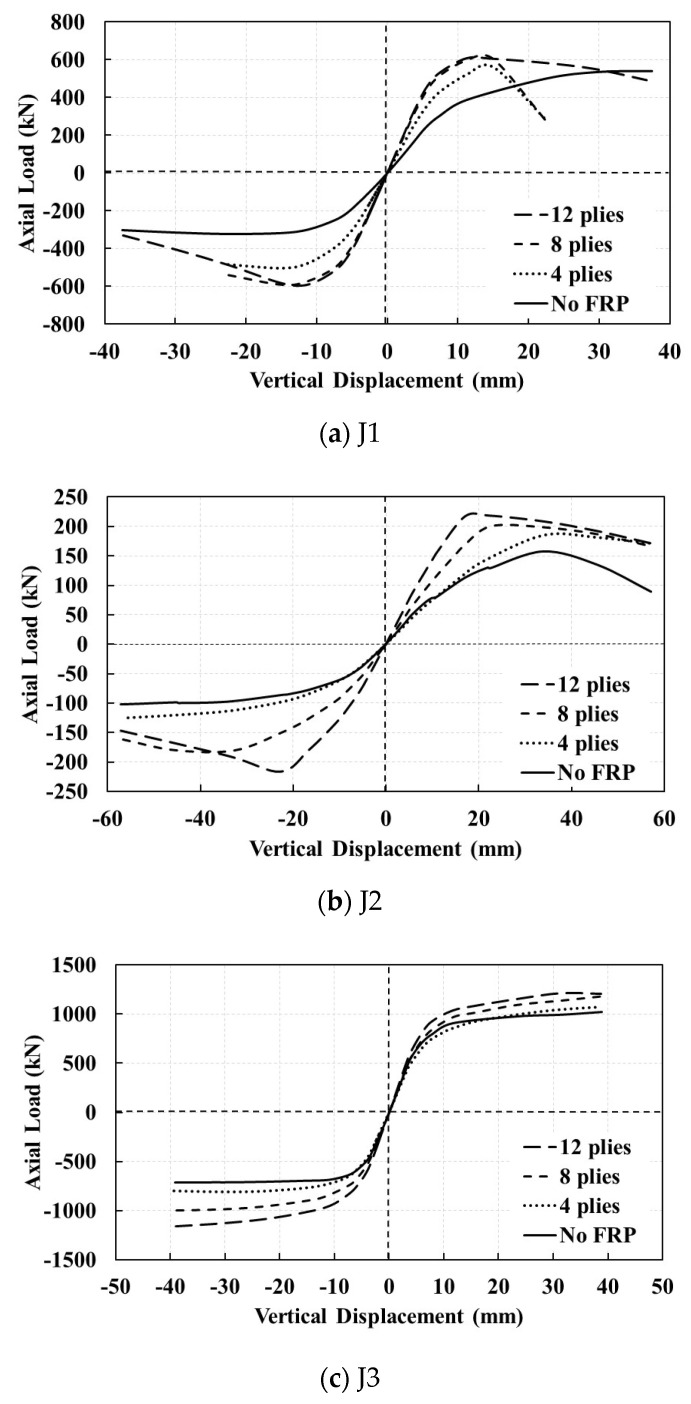
Comparison of the envelope curves of hysteresis load displacement plots with different numbers of FRP layers for both chord and brace: (**a**) J1, (**b**) J2, and (**c**) J3 model.

**Figure 13 materials-14-00648-f013:**
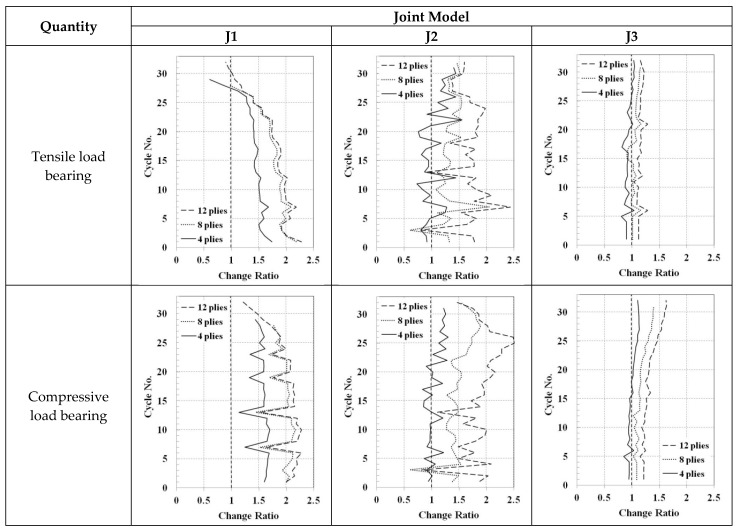
The load bearing capacity and the energy dissipation in each cycle of the cyclic loading for the chord and the brace-retrofitted joints. The values are normalized with respect to the corresponding values of the non-retrofitted case.

**Table 1 materials-14-00648-t001:** Geometric properties of the CHS T-joints studied in this research.

Joint Name	D (mm)	d (mm)	T (mm)	t (mm)	L (mm)	*β*	γ
J1	245	121	12	8	1500	0.49	10.21
J2	458	165	5	5	2286	0.36	45.80
J3	508	406	8	8	1575	0.80	31.75

**Table 2 materials-14-00648-t002:** Thermal and mechanical properties of steel.

Temperature (°C)	Specific Heat (J/g °C)	Conductivity (J/mm °C s)	Density (g/mm^3^)	Yield Stress (Mpa)	Thermal Expansion Coefficient (°C^−1^)	Young’s Modulus (GPa)
0	0.462	0.015	0.790	375	1.70 × 10^−5^	198.5
100	0.496	0.015	0.788	311	1.74 × 10^−5^	193
200	0.512	0.016	0.783	251.14	1.80 × 10^−5^	185
300	0.525	0.018	0.779	231.96	1.86 × 10^−5^	176
400	0.54	0.018	0.775	212.78	1.91 × 10^−5^	167
600	0.577	0.021	0.766	174.42	1.96 × 10^−5^	159
800	0.604	0.024	0.756	136.06	2.02 × 10^−5^	151
1200	0.676	0.032	0.737	59.34	2.07 × 10^−5^	60
1300	0.692	0.034	0.732	40.16	2.11 × 10^−5^	20
1500	0.700	0.120	0.732	1.795	2.16 × 10^−5^	10

**Table 3 materials-14-00648-t003:** Properties of FRP composite material.

Composite Type	E_1_ (MPa)	E_2_ (MPa)	υ_12_	G_12_ (MPa)	G_23_ (MPa)	G_13_ (MPa)
Carbon epoxy	230,000	14,816	0.17	7408	7408	2963

## Data Availability

The data presented in this study are available on request from the corresponding author.

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
