# Peer review of "Numerical Analysis of Axial Cyclic Behavior of FRP Retrofitted CHS Joints"

_materials, 2021, doi:10.3390/ma14030648_

Round 1
Reviewer 1 Report
The paper study numerically the axial cyclic behavior of the circular hollow section T-joints using the finite element method. The residual stresses and strain due to welding process were considered. Three joints with different ratios of chord to brace radius were adopted and the effects of FRP retrofitting on their cyclic behavior were examined. From the numerical results it was shown that the welding process can change initial imperfection of the joints and may affect the buckling modes of the joints under compressive loading. The FRP wrapping can effectively improve the performance of tubular T-joints and can change the buckling mode under cyclic loading.
The paper addresses a topic posing numerical challenges and having practical significance. It is methodologically correct. The paper is suitable for publication.
The literature review in introduction is thorough and it is very well written, however some additional references listing below regarding the extraction develop of fragility curves and structural control of steel structures with tubular T-joints could be added in the introduction.
- Nikos G. Pnevmatikos, George A. Papagiannopoulos, George S. Papavasileiou, (2019), Fragility curves for mixed concrete/steel frames subjected to seismic excitation, Soil Dynamics and Earthquake Engineering 116, 709-713
- Nikos G. Pnevmatikos, “New strategy for controlling structures collapse against earthquakes”. Natural Science, Vol.4, pp. 667-676, 2012, doi:10.4236/ns.2012.428088.
- .
Author Response
The comments of the respected reviewer and your positive stand-point about the manuscript are appreciated. Below are our responses to your comments:
- Comment: “The literature review in introduction is thorough and it is very well written, however some additional references listing below regarding the extraction develop of fragility curves and structural control of steel structures with tubular T-joints could be added in the introduction.
- Nikos G. Pnevmatikos, George A. Papagiannopoulos, George S. Papavasileiou, (2019), Fragility curves for mixed concrete/steel frames subjected to seismic excitation, Soil Dynamics and Earthquake Engineering 116, 709-713
- Nikos G. Pnevmatikos, “New strategy for controlling structures collapse against earthquakes”. Natural Science, Vol.4, pp. 667-676, 2012, doi:10.4236/ns.2012.428088.”
Response: The proposed references were added to the manuscript in Page 1, and its reference list.
We appreciate precise, credible and faithful review of the paper by the respected reviewer.
Reviewer 2 Report
This paper numerically investigates the cyclic behaviour of retrofitted and non-retrofitted circular hollow section T-joints under axial loading. The effects of welding process on buckling instability of the joints in compression, and plastic failure in tension are considered. However, to accept this paper for publication the following comments need to be addressed
- The proficiency of the language needs a more improvement in the manuscript.
- The abstract needs to be rewritten. The results show that the welding process may significantly increase the plastic failure potential. Are you not sure of your claim??
- The introduction section needs to be improved by citing new and related articles of the current journal.
- The Conclusion should be more concise.
Otherwise, the methods and results are quite presented. The manuscript can be accepted after minor revision
Author Response
The comments of the respected reviewer and your positive stand-point about the manuscript are appreciated. Below are our responses to your comments:
- Comment: “The proficiency of the language needs a more improvement in the manuscript.”
Response: The English language of the text was totally reviewed and modified for more clarity and coherency.
- Comment: “The abstract needs to be rewritten. The results show that the welding process may significantly increase the plastic failure potential. Are you not sure of your claim??”
Response: The Abstract was rewritten to be more concise.
- Comment: “The introduction section needs to be improved by citing new and related articles of the current journal.”
Response: New and related articles of the current journal were added to the text and cited in the reference list.
- Comment: “The Conclusion should be more concise.”
Response: The conclusions section was totally reviewed to be more concise.
We appreciate precise, credible and faithful review of the paper by the respected reviewer.
Reviewer 3 Report
The paper proposes a study on the numerical analysis of axial cyclic behavior of FRP retrofitted CHS joints. The paper is clear and concise and it addresses an interesting and relevant problem. The English is acceptable, the figures and tables are well organized. The experimental tests developed in the manuscript are a good contribution to the field, but the Authors should explain some of their choices. The numerical part needs to be expanded since a fundamental part of the manuscript. Finally, I suggest a major revision. In what follows, I list some comments and suggestions that can be addressed by the authors while finalizing the manuscript in a major revision process.
Line 33: An important feature of FRP composite material is its reversibility for certain types of reinforced structure. Please add the following paper: Cascardi, A., Dell'Anna, R., Micelli, F., Lionetto, F., Aiello, M. A., & Maffezzoli, A. (2019). Reversible techniques for FRP-confinement of masonry columns. Construction and Building Materials, 225, 415-428.
Line 60: The authors are invited to expand the bibliography, especially the numerical section since a large part of the paper is based on this type of analysis. The following papers are also recommended for consideration: 1) Chen, G. M., J. G. Teng, J. F. Chen, and Q. C. Xiao. 2015. “Finite element
modeling of the debonding failures in FRP-strengthened RC beams: A
dynamic approach.” Comput. Struct. 158 (Oct): 167–183. https://doi
.org/10.1016/j.compstruc.2015.05.023. 2) Ombres, L.; Verre, S. Experimental and Numerical Investigation on the Steel Reinforced Grout (SRG) Composite-to-Concrete Bond. J. Compos. Sci. 2020, 4, 182.
Line 118: Is the number of cycles imposed by the SAC loading protocol?
Table 3: Have the values reported in Table 3 been experimentally evaluated?
Line 132: In addition, authors should add information about the type of connection adopted (i.e., perfect bond), the materials model adopted and the type of analysis used (dynamic or static).
Line 134: Please report more information about the type of mesh adopted for the J1 models, especially in the connecting section. Also, is the type of mesh adopted irrelevant to the final result?
Figure 5: A color figure is more exhaustive
Figure 7: A color figure is more exhaustive
Figure 11: A color figure is more exhaustive
Figure 12: Authors are invited to increase the quality of Figure 12.
Author Response
The comments of the respected reviewer and your positive stand-point about the manuscript are appreciated. Below are our responses to your comments:
- Comment: “Line 33: An important feature of FRP composite material is its reversibility for certain types of reinforced structure. Please add the following paper: Cascardi, A., Dell'Anna, R., Micelli, F., Lionetto, F., Aiello, M. A., & Maffezzoli, A. (2019). Reversible techniques for FRP-confinement of masonry columns. Construction and Building Materials, 225, 415-428.”
Response: The suggested reference was added to the text in Page 1 and cited in the reference list.
- Comment: “Line 60: The authors are invited to expand the bibliography, especially the numerical section since a large part of the paper is based on this type of analysis. The following papers are also recommended for consideration: 1) Chen, G. M., J. G. Teng, J. F. Chen, and Q. C. Xiao. 2015. “Finite element modeling of the debonding failures in FRP-strengthened RC beams: A dynamic approach.” Comput. Struct. 158 (Oct): 167–183. https://doi
.org/10.1016/j.compstruc.2015.05.023. 2) Ombres, L.; Verre, S. Experimental and Numerical Investigation on the Steel Reinforced Grout (SRG) Composite-to-Concrete Bond. Compos. Sci.2020, 4, 182.”
Response: The suggested papers were added to expand the bibliography in Section 1. Other scientific papers were also added for this purpose.
- Comment: “Line 118: Is the number of cycles imposed by the SAC loading protocol?”
Response: Yes, the number of cycles is proposed by SAC loading protocol.
- Comment: “Table 3: Have the values reported in Table 3 been experimentally evaluated?”
Response: These properties were extracted from a reference (review paper) which was added to the text in Page 5 and cited in the reference list.
- Comment: “Line 132: In addition, authors should add information about the type of connection adopted (i.e., perfect bond), the materials model adopted and the type of analysis used (dynamic or static).”
Response: Proper explanation regarding the issues raised was added to the text in Page 5.
- Comment: “Line 134: Please report more information about the type of mesh adopted for the J1 models, especially in the connecting section. Also, is the type of mesh adopted irrelevant to the final result?”
Response: More information was reported in Page 5 covering this comment.
- Comment: “Figure 5: A color figure is more exhaustive”
Response: Unfortunately, we do not have access to the color version of this figure.
- Comment: “Figure 7: A color figure is more exhaustive”
Response: Unfortunately, we do not have access to the color version of this figure.
- Comment: “Figure 11: A color figure is more exhaustive”
Response: Unfortunately, we do not have access to the color version of this figure.
- Comment: “Figure 12: Authors are invited to increase the quality of Figure 12.”
Response: Figure 12 was replaced with more quality images.
We appreciate precise, credible and faithful review of the paper by the respected reviewer.
Round 2
Reviewer 3 Report
The manuscript can be published in the present form